# Potential Blood Biomarkers for Diagnosing Periprosthetic Joint Infection: A Single-Center, Retrospective Study

**DOI:** 10.3390/antibiotics11040505

**Published:** 2022-04-11

**Authors:** Hong Xu, Jinwei Xie, Shaoyun Zhang, Duan Wang, Zeyu Huang, Zongke Zhou

**Affiliations:** 1Department of Orthopedics, Orthopedic Research Institute, West China Hospital, Sichuan University, Chengdu 610000, China; xuhongvip@yeah.net (H.X.); rain_xjw@yeah.net (J.X.); zhangsyhm2068@163.com (S.Z.); wangduan_bone@163.com (D.W.); zey.huang@gmail.com (Z.H.); 2Department of Orthopedics, The Third Hospital of Mianyang, Sichuan Mental Health Center, Mianyang 621000, China

**Keywords:** periprosthetic joint infection, diagnosis, C-reactive protein, blood markers, revision arthroplasty

## Abstract

Background: Blood biomarkers are first-line tools for identifying periprosthetic joint infection (PJI). C-reactive protein (CRP) is currently recognized as the standard biomarker for PJI diagnosis. Other recently reported novel biomarkers, including plasma fibrinogen, platelet count, monocyte/lymphocyte ratio (MLR), neutrophil/lymphocyte ratio (NLR), and platelet count/lymphocyte ratio (PLR), have also shown promise in diagnosing PJI. This study aimed to evaluate whether these biomarkers were superior to CRP for identifying PJI. Methods: Patients who underwent revision hip or knee arthroplasty at our hospital from January 2008 to September 2020 were included consecutively and divided into infected and non-infected groups according to the 2013 International Consensus Meeting Criteria. Blood samples were collected preoperatively, and erythrocyte sedimentation rate (ESR), CRP, interleukin-6, fibrinogen, platelet count, MLR, NLR, and PLR were analyzed. The diagnostic values of the tested biomarkers and their combinations were compared with CRP based on the area under the receiver operating characteristic curve (AUC) using the z-test. Classification trees were constructed to explore more accurate combinations of the tested markers for identifying PJI. Results: A total of 543 patients were included, of whom 245 had PJI. Among the tested biomarkers, CRP with a cutoff of 7.39 mg/L showed the highest AUC, which gave a sensitivity of 79.1% and specificity of 86.0%. The AUCs of pairwise combinations of tested markers including CRP also were inferior to CRP itself, as were combinations derived from classification trees. Conclusions: Preoperative serum CRP with a low cutoff may be the best reliable blood biomarker for identifying PJI, and those traditional or novel available blood biomarkers could not further improve the diagnostic ability on the basis of CRP.

## 1. Introduction

Prosthetic joint infection (PJI) is a catastrophic complication after total joint arthroplasty (TJA), which is associated with higher hospitalization costs, longer hospital length of stay, and even higher disability and mortality rates [1,2]. Although the incidence of PJI is lower than 2%, the rate of PJI is expected to increase rapidly with the increasing number of primary TJA in the next 30 years [3,4]. Hence, a reliable method to identify PJI among TJA patients is needed. However, the identification of infection before revision arthroplasty still poses a great challenge to surgeons. Timely and accurate preoperative identification of PJI would inform treatment planning, management of patient expectations, and decision-making about whether or not to preserve the implanted prosthesis [5,6].

Although the discovery of some biomarkers such as synovial alpha-defensin [7] and the development of next-generation sequencing [8] have improved the potential for preoperative detection of PJI, these techniques are expensive and inconvenient, especially for outpatients and for smaller hospitals. In contrast, blood-based biomarkers may be more suitable as first-line tools for identifying PJI, given their easy accessibility, cost-effectiveness and rapid determination [9]. Serum C-reactive protein (CRP), D-dimer and erythrocyte sedimentation rate (ESR) are the blood biomarkers currently recommended by International Consensus Meeting (ICM) in 2018 to identify PJI [10], while the European Bone and Joint Infection Society recommends only CRP in their latest guidelines [11].

Recently, some studies have reported that several novel blood biomarkers, such as serum D-dimer [12], plasma D-dimer [13], plasma fibrinogen [9,14,15], calprotectin [16], platelet count and mean platelet volume, monocyte/lymphocyte ratio (MLR), neutrophil/lymphocyte ratio (NLR), and platelet count/lymphocyte ratio (PLR) may be helpful for identifying PJI [17,18,19]. However, studies have come to different conclusions about the optimal biomarkers or their combinations for identifying PJI. Moreover, some of those studies [9,13,15,19] determined superior diagnostic markers by simply comparing the absolute values of the areas under the receiver operating characteristic curves (AUCs) instead of performing the z-test on the AUCs, and the statistical defects may lead to inappropriate results and conclusions. It is important to determine how these biomarkers, or their combinations, compare with the widely used serum CRP. Furthermore, determining the optimal biomarkers, alone or in combination, is also important because if a combination works better, it may be more complex to implement in the clinic.

With these goals in mind, we performed the present single-center, retrospective study to (1) evaluate whether certain blood biomarkers individually or in pairwise combinations could diagnose PJI better than serum CRP on its own; and (2) explore whether certain combinations of tested markers were superior to CRP based on classification trees.

## 2. Materials and Methods

### 2.1. Study Design

Patients who underwent revision hip or knee arthroplasty from January 2008 to September 2020 at our hospital were included retrospectively in this study. The Ethics Committee of the hospital approved our study (approval no. 2020-1004). It also waived the requirement for written informed consent since this was a retrospective study, it could not cause any adverse effects for included patients, and the patient data were reported anonymously. This study was registered in the Chinese Clinical Trial Registry (approval no. 2020-1004).

### 2.2. Inclusion and Exclusion Criteria

Patients who underwent revision hip or knee arthroplasty were identified according to the procedure codes of the Clinical Modification of the International Classification of Diseases (10th revision) [20] and were consecutively included. Initially, a total of 743 patients were enrolled. Among them, 65 patients who underwent revision arthroplasty for periprosthetic fracture and 7 for hip joint dislocation were excluded because they were caused by violence or surgical technique instead of PJI. Another 157 patients who underwent hip or knee reimplantation surgery were excluded because the duration of infection and source of responsible pathogens were uncertain [14,21]. Finally, 543 patients were included in our analysis, composed of 245 patients with diagnosed infection and 298 without.

### 2.3. Diagnostic Definition of Infection and Data Extraction

PJI in our study was defined according to the 2013 ICM PJI Criteria [22]. The following data were extracted from our hospital’s electronic medical records as reported [14]: demographic information such as sex, age, and body mass index; diagnoses and comorbidities, including hypertension, type 2 diabetes, chronic obstructive pulmonary disease, coronary heart disease, and inflammatory diseases such as rheumatoid arthritis and ankylosing spondylitis; results of preoperative laboratory tests; pathology results of soft tissue around the implant, collected intraoperatively; and the results of pathogenic cultures of synovial fluid aspirated before and during surgery.

### 2.4. Laboratory Evaluations of Tested Markers

Fasting venous blood samples for all patients were collected by nurses on the day of admission, then the blood samples were sent to our hospital’s department of laboratory medicine and tested in a timely manner as described [14]. The tested biomarkers included ESR, serum CRP, interleukin-6 (IL-6), and plasma fibrinogen, as well as the routine blood indices of platelet count, and monocyte, lymphocyte, and neutrophil counts. If PJI was suspected based on clinical signs such as pain and swelling, and the results of preoperative CRP and ESR, the involved hip or knee joint was aspirated on the day of admission or the first day after admission, and the obtained synovial fluid was sent for testing in a timely manner, which included white blood cell count, neutrophil count, and polymorphonuclear neutrophil percentage, as well as aerobic and anaerobic cultures with blood culture bottle. Moreover, the synovial fluid of all patients was collected during surgery and sent for testing. In addition, at least four soft tissues around the prosthesis were collected and sent for culture and histologic tests intraoperatively. Furthermore, the definition of positive histology was: >5 neutrophils per high-power field in 5 high-power fields (×400).

### 2.5. Statistical Analysis

Our sample size was considered adequate for evaluating the tested biomarkers for their ability to diagnose PJI, based on previous work [23]. The categorical variables were presented as frequency (percentage) and assessed for significance using Pearson chi-squared test and Fisher’s exact test for two groups. Normally distributed continuous variables were presented as mean (standard deviation, SD) and assessed using Student’s t test, while skewed data were described as median (interquartile range) and assessed using the Mann-Whitney U test.

Receiver operating characteristic curves were constructed and employed to describe the relationships between the true-positive rate (sensitivity) and false-positive rate (1-specificity) and calculate AUCs together with their 95% confidence intervals (CIs) for tested biomarkers or their combinations, which included CRP with other biomarkers. The z-test was used to compare AUCs for tested biomarkers, alone or in combination, against the AUC for CRP. Positive predictive value (PPV) and negative predictive value (NPV) were also calculated. Optimal predictive cutoffs for all tested biomarkers were determined based on the Youden index. Classification trees were constructed to explore more reliable combinations of the tested biomarkers. Classification trees pick the distinguishing variable and give the best discriminatory cutoff value at each parent node. The internally cross-validated repeated 100 times to get the optimal cutoff. And chi-square *p* values < 0.05 for the distribution at break points. In consideration of the clinical practicality, tree depth was set as 1–3. All statistical analysis was performed using SPSS 24 (IBM, Armonk, NY, USA). Differences associated with *p* < 0.05 were considered statistically significant.

## 3. Results

A total of 543 patients who underwent revision hip or knee arthroplasty were included in our study, of whom 245 were diagnosed with infection (Figure 1). The demographic characteristics and comorbidities of infected and uninfected patients are summarized in Table 1. All biomarkers in our study were tested preoperatively except for IL-6, which was not routinely tested preoperatively in our department during the study period. A total of 149 patients with infection and 215 without infection were tested for IL-6. In addition, the culture results of the infected patients were showed in Appendix A, *Staphylococcus epidermidis* and *Staphylococcus aureus* were more than 50%.

The levels of all tested blood biomarkers were significantly higher in the infected group than in the non-infected group (*p* < 0.001; Table 2). Firstly, we assessed the diagnostic ability of the eight tested biomarkers on their own for identifying PJI. Our results showed that serum CRP had the highest AUC (0.882, 95% CI 0.846–0.918), which gave a sensitivity of 79.1% and a specificity of 86.0% with the optimal predictive cutoff of 7.39 mg/L. Moreover, the AUCs of other tested biomarkers were significantly lower than that of CRP (Figure 2a and Table 3).

Secondly, we systematically evaluated different pairwise combinations of the tested markers to explore whether they might out-perform CRP for PJI diagnosis. Our results revealed that the diagnostic abilities of all pairwise combinations of the tested markers remained inferior to that of CRP (Figure 2b–f and Appendix A).

Finally, we constructed classification trees with different tree depths to explore more reliable combinations for identifying PJI with the tested biomarkers. CRP was enrolled in the classification tree with an accuracy of 82.5% if the tree depth was set as one (Appendix A). CRP and ESR were enrolled if the tree depth were set as two, which gave an accuracy of 81.8% (Appendix A). CRP, ESR, and PLR were successively enrolled if the tree depth was set as three, which gave an accuracy of 62.6% (Appendix A and Table 4). These results suggested that classification trees also could not provide more reliable combinations for identifying infection.

## 4. Discussion

Our study suggests that both traditional inflammatory blood biomarkers including ESR and IL-6, as well as several recently reported novel blood biomarkers such as fibrinogen, platelet count, MLR, NLR, and PLR, alone or in combination, fail to diagnose PJI more accurately than serum CRP. Furthermore, CRP with a low cutoff may be the best reliable blood marker for identifying PJI.

Early and accurate identification of PJI is a prerequisite for effective treatment [11]. Blood biomarkers play important roles for identifying PJI because they are easily accessible and highly cost-effective. Numerous scholars have tried their best to find more accurate blood biomarkers on the basis of established blood biomarkers so as to improve the diagnostic ability for PJI. However, the optimal blood biomarker(s) for diagnosing PJI have yet to be determined. More importantly, though several novel blood biomarkers, such as fibrinogen, platelet count, MLR, NLR, and PLR, have been reported that may promising for diagnose of PJI, it is necessary to evaluate whether they can further improve the diagnostic ability on the basis of the classic inflammatory biomarker, CRP. In addition, ICM recommended serum D-dimer, CRP, and ESR in 2018 [10], while European Bone and Joint Infection Society only recommended CRP in their latest guideline [11] for identifying PJI. ESR testing was insensitive for identifying infection, our study also showed that ESR could not also improve the accuracy for identifying PJI on the basis of CRP, whether it is used alone or combined with other blood markers, which is in line with previous studies [18,19]. In addition, our study did not analyze serum D-dimer for identifying PJI since only plasma D-dimer was tested in our hospital. Furthermore, the differences between the two D-dimer measurements are unclear [9]. Although some studies [9,13,24,25] have suggested that plasma D-dimer has limited ability in identifying PJI, we believe further studies comparing the diagnostic values of plasma and serum D-dimer levels in the same patients are needed.

Fibrinogen, a large (340 kDa) hexameric homodimer secreted by the liver, plays a vital role in hemostasis and homeostasis [26]. Several studies [9,14] suggested that plasma fibrinogen was a promising biomarker for identifying PJI, and one study [27] even concluded that its diagnostic value was higher than that of serum CRP. However, our study suggests that its diagnostic ability remains inferior to that of serum CRP, whether it is used alone or combined with other blood biomarkers. This result is consistent with previous work by Wu et al. [28]. Our results may be more reliable than many previous studies [9,14,27,28] because those studies did not perform the z-test to compare the AUCs of plasma fibrinogen and serum CRP. Not applying the z-test can cause misleading results, especially when the sample size is small. Nevertheless, previous work by Xu et al. showed that plasma fibrinogen may be useful for identifying PJI in patients with inflammatory diseases [14], which should be verified in larger studies.

IL-6, a cytokine secreted by macrophages, monocytes, and T cells after these cells are activated under inflammation, plays an important role in autoimmune diseases and infection [29]. One study showed that serum IL-6 was able to diagnose PJI with a sensitivity of 100% and a specificity of 90.9% based on a cutoff of >10.4 pg/mL, but only 11 infected patients were included in the study. A meta-analysis [30] involving nine studies showed that serum IL-6 gave a sensitivity of 72% and higher specificity of 89% for identifying PJI, which is near to the specificity (70.9%) and sensitivity (86.5%) in the present study. Even so, IL-6 did not diagnose PJI more accurately than CRP in our study.

MLR and NLR were first reported to be useful for diagnosing community-acquired pneumonia [31]. Increases in MLR and NLR may be associated with the fact that monocyte and neutrophil counts usually increase during bacterial infection, whereas lymphocyte count usually decreases. However, our study revealed that both MLR and NLR failed to diagnose PJI more accurately than CRP. In contrast, one study [19] showed that the AUC of NLR was slightly higher than that of serum CRP (0.802 vs. 0.793) for the diagnosis of early PJI, but that study did not compare their AUCs using a z-test, and this small difference may not be statistically or clinically significant. Hence, further studies are needed to further evaluate the diagnostic ability of NLR for identifying early PJI.

Platelet-related biomarkers, such as platelet count, PLR, and the ratio of platelet count to mean platelet volume, have been explored for identifying PJI [17,18]. Our analysis also suggests that platelet count and PLR, alone or combined with other blood markers, do not diagnose PJI better than serum CRP. Although we did not evaluate the platelet count/volume ratio in the present study, since mean platelet volume was not routinely tested in our blood analyses, one study [17] showed that platelet count/mean platelet volume ratio provided a poor diagnostic value for identifying PJI with a low sensitivity of 48.1% and a specificity of 80.9%.

Our study revealed that serum CRP has the highest accuracy for identifying PJI among tested blood biomarkers. However, the optimal cutoff generated from the Youden index in the present study (7.39 mg/L) is substantially lower than that recommended by the 2013 ICM criteria [22], as well as the European Bone and Joint Infection Society (10 mg/L) [31]. To be the first-line biomarker, lower CRP cutoff could increase the sensitivity and reduce the rate of missed PJI diagnoses [32]. One study [33] reported that a cutoff of 10 mg/L gave unsatisfying sensitivity, while another [34] found that a cutoff of 5 mg/L gave a sensitivity higher than 95%. Hence, further study may be necessary to optimize the CRP cutoff.

Some limitations in this study should be mentioned. First, although our sample size is sufficient, they are all from one center, and the test method and time of delivery of samples to test may affect our results. Therefore, studies from other centers are needed. Second, the patients in our study were from same race. It is still unclear whether races affect the tested biomarkers; hence, studies based on other races are needed. Finally, the optimal cutoff of serum CRP should be determined in future study on the basis of balancing sensitivity and specificity simultaneously.

Although our study showed that the tested blood biomarkers could not diagnose PJI in our sample more accurately than serum CRP according to the z-test of AUCs and classification tree, some biomarkers such as NLR may still be useful for identifying early PJI [19], and plasma fibrinogen may aid diagnosis of PJI in patients with inflammatory diseases [14]. In addition, fibrinogen or IL-6 may be useful for diagnosing PJI in patients when data on CRP are absent.

## 5. Conclusions

Among traditional inflammatory blood biomarkers and several novel blood biomarkers recently reported, preoperative serum CRP with a low cutoff may be the most reliable for identifying PJI, and others, no matter alone or in combination, could not further improve the diagnostic ability on the basis of CRP.

A total of 543 patients who underwent revision hip or knee arthroplasty were included in our study, of whom 245 were diagnosed with infection.

Serum CRP had the highest AUC among the tested markers individually, and all pairwise combinations of the tested markers remained inferior to that of CRP.

## Figures and Tables

**Figure 1 antibiotics-11-00505-f001:**
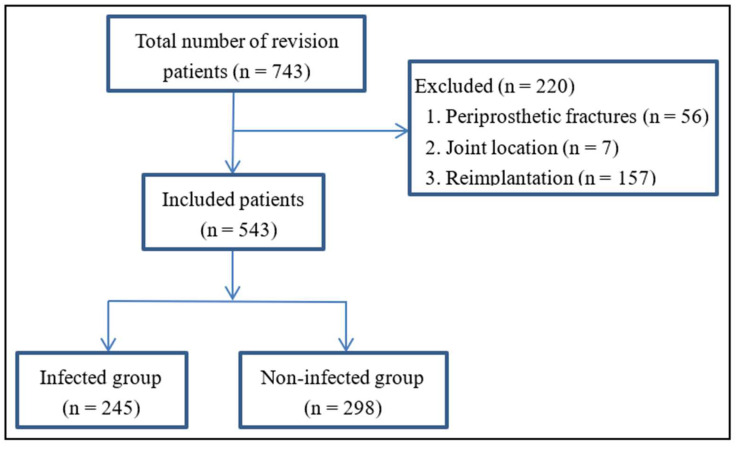
Flow diagram of the study design.

**Figure 2 antibiotics-11-00505-f002:**
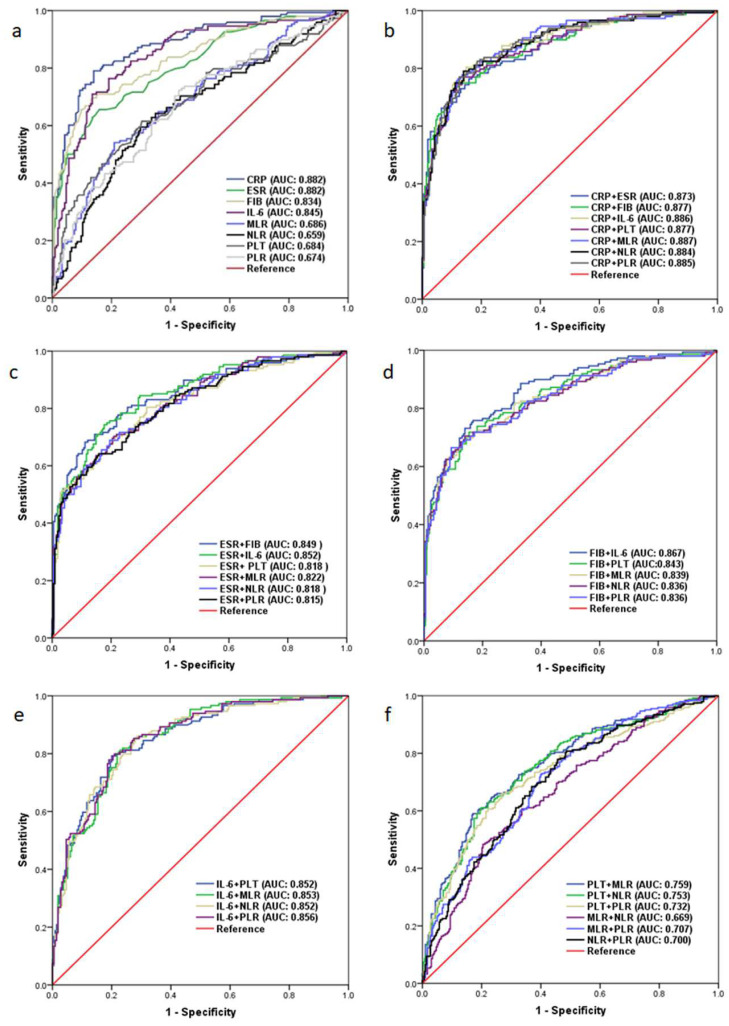
Receiver operating characteristic curves of the tested biomarkers. (**a**) C-reactive protein (CRP), erythrocyte sedimentation rate (ESR), fibrinogen (FIB), interleukin-6 (IL-6), platelet count (PLT), monocyte/lymphocyte ratio (MLR), neutrophil/lymphocyte ratio (NLR), and platelet count/lymphocyte ratio (PLR) on their own. (**b**) Combinations of CRP with other markers. (**c**) Combinations of ESR with other markers. (**d**) Combinations of FIB with other markers. (**e**) Combinations of IL-6 with other markers. (**f**) Combinations of PLT, MLR, and NLR with each other.

**Table 1 antibiotics-11-00505-t001:** Characteristics of patients with or without periprosthetic joint infection.

Variable	Infected Group (n = 245)	Non-Infected Group (n = 298)	*p*-Value
Demographic characteristics			
Age, yr	61.64 ± 13.56	63.90 ± 10.70	0.059
Female	117 (47.8)	180 (60.4)	0.003 *
BMI (kg/m^2^)	23.47 ± 3.47	23.91 ± 3.45	0.273
Comorbidities			
Hypertension	60 (24.5)	112 (37.6)	0.001 *
Diabetes	29 (11.8)	23 (7.7)	0.105
COPD	3 (1.2)	4 (1.3)	0.904
CHD	3 (1.2)	11 (3.7)	0.071
Inflammatory diseases	18 (7.3)	15 (5.0)	0.262

Values were presented as n (%) or mean ± SD, unless otherwise noted. * *p* < 0.05. BMI: Body Mass Index; COPD: chronic obstructive pulmonary disease; CHD: coronary heart disease.

**Table 2 antibiotics-11-00505-t002:** Tested biomarkers in the infected and non-infected groups.

Potential Biomarker	Infected Group (n = 245)	Non-Infected Group (n = 298)	*p-*Value
CRP (mg/L)	19.30 (10.40–44.30) ^#^	3.07 (1.98–5.32) ^#^	<0.001 *
ESR (mm/h)	61.54 ± 32.68	26.46 ± 18.95	<0.001 *
FIB (g/L)	4.41 ± 1.33	2.99 ± 0.77	<0.001 *
IL-6 (pg/mL)	14.21 (7.07–27.52) ^#^	3.56 (2.16–5.89) ^#^	<0.001 *
PLT (×10^9^/L)	242.94 ± 92.23	178 ± 65.30	<0.001 *
MLR	0.33 ± 0.19	0.25 ± 0.14	<0.001 *
NLR	3.21 (2.22–4.43) ^#^	2.34 (1.75–3.15) ^#^	<0.001 *
PLR	178.90 ± 92.09	128.21 ± 66.04	<0.001 *

Values were presented as mean ± SD, unless otherwise noted. * *p* < 0.05. ^#^ Data were presented as median (interquartile range). CRP: C-reactive protein; ESR: erythrocyte sedimentation rate; FIB: fibrinogen; IL-6: interleukin-6; PLT: platelet count; MLR: monocyte/lymphocyte ratio; NLR: neutrophil/lymphocyte ratio; PLR: platelet count/lymphocyte ratio.

**Table 3 antibiotics-11-00505-t003:** Diagnostic performance of the tested biomarkers individually.

Biomarker	AUC (95% CI)	Youden Index	Optimal Cutoff	Sensitivity	Specificity	PPV	NPV	*p*-Value Compared with CRP
CRP (mg/L)	0.882 (0.846–0.918)	0.651	7.39	79.1%	86.0%	82.3%	83.4%	-
ESR (mm/h)	0.809 (0.736–0.885)	0.414	42.5	65.5%	84.2%	77.3%	74.8%	<0.001 *
FIB (g/L)	0.834 (0.791–0.878)	0.561	3.67	69.6%	86.5%	80.9%	77.6%	<0.001 *
IL-6 (pg/mL)	0.845 (0.803–0.887)	0.574	8.59	70.9%	86.5%	81.2%	78.3%	0.038 *
PLT (×109/L)	0.684 (0.626–0.741)	0.313	201.5	61.5%	69.8%	62.6%	68.8%	<0.001 *
MLR	0.686 (0.630–0.742)	0.332	0.30	54.1%	79.1%	68.0%	67.7%	<0.001 *
NLR	0.659 (0.601–0.717)	0.293	2.90	62.8%	66.5%	60.7%	68.5%	<0.001 *
PLR	0.674 (0.617–0.731)	0.295	126.11	72.3%	57.1%	58.1%	71.5%	<0.001 *

* *p* < 0.05. AUC: area under the receiver operating characteristic curve; 95% CI: 95% confidence interval; CRP: C-reactive protein; ESR: erythrocyte sedimentation rate; FIB: fibrinogen; IL-6: interleukin-6; PLT: platelet count; MLR: monocyte/lymphocyte ratio; NLR: neutrophil/lymphocyte ratio; PLR: platelet count/lymphocyte ratio; PPV: positive predictive value; NPV: negative predictive value.

**Table 4 antibiotics-11-00505-t004:** Diagnostic performance of the tested biomarkers based on the classification tree.

Tree Depth	Enrolled Marker and Cutoff	Sensitivity	Specificity	PPV	NPV	Accuracy
1	CRP (5.91 mg/L)	86.1%	79.5%	77.57%	87.5%	82.5%
2	CRP (5.91 mg/L) + ESR (32 mm/h)	71.8%	89.9%	85.4%	79.5%	81.8%
3	CRP (5.91 mg/L) + ESR (32 mm/h) + PLR (131.80)	58.0%	66.4%	58.7%	65.8%	62.6%

CRP: C-reactive protein; ESR: erythrocyte sedimentation rate; PLR: platelet/lymphocyte ratio; PPV: positive predictive value; NPV: negative predictive value.

## Data Availability

The datasets used and analyzed during the current study are available from the first or corresponding author on reasonable request.

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
