# Peer review of "Potential Blood Biomarkers for Diagnosing Periprosthetic Joint Infection: A Single-Center, Retrospective Study"

_antibiotics, 2022, doi:10.3390/antibiotics11040505_

Round 1

Reviewer 1 Report

The manuscript is interesting and the topic is relevant

Some changes are needed:

  • the structure of the manuscript should be changed: materials and methods should go in order before results and discussion
  • - supplementary materials (page 9) with explanation of cutoff values should be included in the main text
  • table 4 should be clearer (maybe some cells or lines); in the captions please correct platele into platelet
  • cutoff values for ESR, CPR and PLR are mentioned; no mention is made for other biomarkers (IL6, fibrinogen,...) why? please include or justify in study design

Author Response

Please see the attachment, thank you!

Reviewer 2 Report

In the article, Potential Blood Biomarkers for Diagnosing Periprosthetic Joint Infection: A Single-Center, Retrospective Study, authors Xu et. al. have studied a few biomarkers to diagnose PJI and compared it with CRP.

The work seems to be very repetitive, there are many-many such articles that have been published in the last few years. Authors could have just googled and found out if there are better biomarkers than CRP. All the studied markers in this study have been previously well stated.

Author Response

Please see the attachment, thank you!

Reviewer 3 Report

The authors described "Potential Blood Biomarkers for Diagnosing Periprosthetic Joint Infection: A Single-Center, Retrospective Study."

They revealed that several commonly used blood biomarkers including ESR, fibrinogen, IL-6, platelet count, MLR, NLR and PLR, alone or in combination, fail to diagnose PJI more accurately than serum CRP. This manuscript is attractive for potential readers especially in Orthopedic surgery. I have some questions to improve this manuscript.

1. In section, "Laboratory Evaluations of Tested Markers", when was the average date you aspirated and examined? What do you mean "acute phase"?

2. How about procalcitonin (PCT) to detect infection?

3. What kind of pathogens were detected? Please add the pathogens in Table.

Author Response

Please see the attachment, thank you!

Round 2

Reviewer 1 Report

Authors improved the manuscript according to Reviewers' requests

Author Response

Thank you very much!

Reviewer 2 Report

 Authors Xu et. al. have made considerable changes in the MS. However, my biggest concern still remains the same. No novelty either in the study or in the conlcusion. In there letter to the reviewer they have explained well that how and why there study is different to other previously pusblished work. However, in the main MS they do not talk about it.  As a reader I want to know whay I should cite your paper or what is the end point conlcusion. The conslusion is same, CRP is the bes biomarker. Like the way it has been used from decades. 

I find it very offensive when authors upload any documnet which is in foregin language and they do not even help reveiweres to understand what it is. I am not sure what is the relevance of non-published document what authors have attached. If atleast they can mention in the letter to reveiw, we could understand. 

I suggest authors to rewrite atleast the conclusion and discussion part. 

Reviewer 3 Report

The authors revised the manuscript precisely. Thank you for this opportunity.

Author Response

Thank you very much!